# Changes in Loneliness, BDNF, and Biological Aging Predict Trajectories in a Blood-Based Epigenetic Measure of Cortical Aging: A Study of Older Black Americans

**DOI:** 10.3390/genes14040842

**Published:** 2023-03-31

**Authors:** Ronald L. Simons, Mei Ling Ong, Man-Kit Lei, Steven R. H. Beach, Yue Zhang, Robert Philibert, Michelle M. Mielke

**Affiliations:** 1Department of Sociology, University of Georgia, Athens, GA 30602, USA; 2Center for Family Research, University of Georgia, Athens, GA 30602, USA; 3Department of Psychology, University of Georgia, Athens, GA 30602, USA; 4Department of Psychiatry, University of Iowa School of Medicine, Iowa City, IA 52242, USA; 5Department of Epidemiology and Prevention, Wake Forest University School of Medicine, Winston-Salem, NC 27101, USA

**Keywords:** cortical aging, brain age, loneliness, BDMF, dementia, African Americans

## Abstract

A recent epigenetic measure of aging has developed based on human cortex tissue. This cortical clock (CC) dramatically outperformed extant blood-based epigenetic clocks in predicting brain age and neurological degeneration. Unfortunately, measures that require brain tissue are of limited utility to investigators striving to identify everyday risk factors for dementia. The present study investigated the utility of using the CpG sites included in the CC to formulate a peripheral blood-based cortical measure of brain age (CC-Bd). To establish the utility of CC-Bd, we used growth curves with individually varying time points and longitudinal data from a sample of 694 aging African Americans. We examined whether three risk factors that have been linked to cognitive decline—loneliness, depression, and BDNFm—predicted CC-Bd after controlling for several factors, including three new-generation epigenetic clocks. Our findings showed that two clocks—DunedinPACE and PoAm—predicted CC-BD, but that increases in loneliness and BDNFm continued to be robust predictors of accelerated CC-Bd even after taking these effects into account. This suggests that CC-Bd is assessing something more than the pan-tissue epigenetic clocks but that, at least in part, brain health is also associated with the general aging of the organism.

## 1. Introduction

Building on the fact that epigenetic modifications are a hallmark of aging, the most promising biomarkers of biological age are the recently developed, second-generation epigenetic measures of accelerated aging, which focus on the prediction of morbidity and mortality. The indices, commonly recognized as epigenetic clocks, offer insights into an individual’s biological aging rate. Empirical research derived from longitudinal data sets reveals the indices to be reliable indicators of the physiological transformations linked to senescence [1,2]. Although there is only minimal overlap in the CpG sites included in the various clocks, they show similar accuracy in predicting morbidity and aging. These clocks are based on loci that predict morbidity and all-cause mortality, suggesting that they reflect changes across tissue types. This might be seen as an indication that some pan-tissue mechanism is driving epigenetic aging [3].

While these pan-tissue clocks are robust predictors of age-related biological changes and all-cause mortality, they are less effective in predicting brain aging and neurological disorders [4]. Indeed, in brain tissue, the clocks tend to under-predict the rate of biological aging and false positives for neurodegenerative disease [5]. Moreover, it is important to note that the accuracy of these clocks tends to decrease in older individuals, particularly during the onset of neurological diseases, which are often associated with advanced age [6]. In an effort to address the limitations of the blood-based epigenetic clocks, Shireby et al. recently developed an epigenetic measure of aging based on human cortex tissue [5]. This cortical clock dramatically outperformed extant blood-based epigenetic clocks in predicting brain age. Further, in subsequent analyses, this cortical clock (CC) showed consistent and robust relationships with biomarkers of Alzheimer’s disease, Lewy body pathology, Parkinson’s disease, and cognitive measures of dementia, whereas several established clocks trained in blood samples (Hannum, Horvath, PhenoAge, and GrimAge) showed small and inconsistent associations with these biomarkers [7].

Unfortunately, a measure based on brain tissue is of limited value to scientists pursuing observational studies to investigate the causes of dementia. Grodstein et al. acknowledge this point and suggest the importance of focusing on CpG sites which are conserved across blood and brain tissue [7]. Consonant with their proposal, the present study strives to investigate whether the CpG sites included in the CC developed by Shireby et al. [5] might be calculated using whole blood (CC-Bd), and used to provide a window on brain aging that goes beyond that provided by second generation indices of accelerated aging known to predict morbidity and mortality (i.e., PhenoAge, GrimAge, and DunedinPACE).

Indeed, by using blood as the tissue source, we anticipate that the precision of the measure as an indicator of brain age would be decreased, but it might still function as a better indicator of brain health than established epigenetic clocks. There are several reasons for optimism in this regard. First, the sites included in Shireby et al.’s [5] cortical clock were selected because of their association with brain aging, and they show almost no overlap with the sites included in the extant epigenetic clocks. Second, based on a small sample of 41 individuals, Grodstein et al. found that CC-Bd showed a modest correlation with the cortical index calculated with brain tissue [8]. Third, in preliminary analyses with the African American sample used in the present study, we found that the inter-correlation between the three established epigenetic clocks included in our analyses (PhenoAge, GrimAge, and DunedinPACE) were stronger than the correlations between CC-Bd and these clocks. This pattern is consistent with the idea that CC-Bd is capturing variance that is unique, relative to the general biological aging reflected in the other clocks. Finally, in a subsample of 186 individuals from the current study sample who completed the Montreal Cognitive Assessment (MoCA), the correlations between CC-Bd and most of the subcomponents of the MoCA were between −0.20 and −0.30, indicating that CC-Bd is related to cognition. Further, these correlations remained after controlling for the three epigenetic clocks—PhenoAge, GrimAge, and Dunedin PACE (see Online Appendix A).

## 2. The Present Study

The present study provides a further test of the extent to which CC-Bd might be functioning as a measure of brain age, independent of other epigenetic clocks. Specifically, we examine whether three risk factors that have previously been linked to cognitive decline and dementia, and that are elevated for older African Americans, predict the acceleration of CC-Bd and do so beyond the effect of established epigenetic clocks. Our analyses control for several variables, such as age, gender, and education, as well as for three new-generation epigenetic clocks. Thus positive results will indicate that our risk factors accelerate CC-Bd beyond any influence of general biological aging.

The three risk factors included in our investigation—psychological depression, loneliness, and methylation of the BDNF gene–pose very different threats to the brain, thereby providing a test of the extent to which a range of cognitive risk factors predict the acceleration of CC-Bd beyond what would be expected based on established epigenetic clocks. The first two factors involve negative emotional states that have been linked to neurobiological changes and cognitive decline [9,10,11], whereas the third factor is a key molecule involved in brain plasticity [12]. Importantly, as discussed below, past research indicates that midlife assessments of each of these factors predict the onset of dementia as much as ten years later. Finally, we selected these particular factors because each of them is potentially subject to intervention. Therefore, to the extent that these risk factors predict the acceleration of CC-Bd, net of the various controls, they might be seen as suggesting avenues for preventive intervention.

### 2.1. Depression

Multiple longitudinal studies have indicated a correlation between midlife depression and the onset of dementia several years later [13,14,15,16], which may be attributed to the neurobiological changes associated with major depressive disorder, including mood disturbances and deficits in memory, attention, and visuospatial processing, which are accompanied by a reduction in hippocampal volume of about 10% according to numerous neuroimaging studies [9]; while some restoration of volume following treatment is possible [17], recurrent depression is likely to be causally linked with the subsequent emergence of dementia. Importantly, given our African American sample, exposure to racism and discrimination has been shown to increase levels of depression [18], effects that may be present from a young age [19].

### 2.2. Loneliness

Human beings have a deeply rooted need for social connection to others [20]. Despite the importance of social relationships for well-being, they tend to diminish with age, leaving older individuals susceptible to experiencing loneliness, a prevalent issue in the US, as approximately 20–30% of older adults report such feelings based on national population estimates [21]. Importantly, loneliness has been shown to increase inflammation, blood pressure, depression, heart disease, and mortality [22].

Recent studies suggest that loneliness is more common among African Americans than white non-Hispanics [23,24]. In large measure, this may be a consequence of racial discrimination and social isolation experienced when interacting in White spaces [25]. Several studies, including longitudinal investigations, have reported that exposure to racial discrimination increases feelings of loneliness [26,27]. This elevated prevalence of loneliness may contribute to African Americans’ higher risk of cognitive decline, as several longitudinal studies have identified loneliness as a risk factor for dementia and Alzheimer’s disease [28,29,30]. These studies include samples from countries all over the world, including China [31], the Netherlands [32], the United Kingdom [33], Singapore [34], and the US [28,29]. Follow-up in most cases ranged from five to ten years, and overall findings held after controlling for social isolation and gender.

Of course, loneliness often leads to depression, making it difficult to distinguish between the two as risk factors. There is, however, a growing consensus that the two are conceptually distinct constructs that can be distinguished clinically [35] and statistically [36,37]. Further, evidence suggests that they have distinctive brain circuitry [10,11,37] and that they exercise independent effects as risk factors for cognitive decline and dementia [28,36,38].

### 2.3. Brain-Derived Neurotropic Factor (BDNF)

Many studies have demonstrated the critical role of BDNF in cognition, especially memory formation and maintenance. It promotes synaptic consolidation and increases neurogenesis through changes in cell survival and proliferation [12]. Changes in BDNF expression are associated with both normal and pathological aging. Reductions in BDNF have been linked to hippocampal and neural shrinkage. Importantly, BDNF protein and mRNA levels are reduced in the post-mortem brain of MCI and AD patients, and in other studies, BDNF levels are correlated with the severity of AD [39]. Further, higher BDNF is associated with a reduced risk of AD/ADRD [40,41], whereas increases in BDNF through pharmacology or exercise is related to better cognitive function and diminished synaptic dysfunction [42,43]. Consonant with such findings, several recent studies have reported that methylation (downregulation) of the promoter region of BDNF (BDNFm) increases the risk for AD/ADRD [44,45,46,47,48].

Several studies have reported that methylation of BDNF is associated with exposure to adversity and trauma across the life course [49,50]. This includes events and circumstances such as childhood stressors, community crime, domestic violence, and combat. Unfortunately, the everyday social conditions of contemporary society elevate the probability that African Americans will experience such difficulties and dangers.

### 2.4. Study Hypotheses

Based on these findings, we expect that over an 11-year period, increased depression, loneliness, and *BDNFm* will each evince an independent effect on the acceleration of CC-Bd and that these effects will remain robust after controlling for age, gender, education, and general epigenetic aging. We will control for epigenetic aging using new-generation epigenetic clocks that have been trained on blood-based biomarkers of physiological decline in addition to chronological age. These clocks have been shown to predict morbidity and mortality, and in some cases, dementia.

The first epigenetic clock, PhenoAge [51,52], has been shown to predict biomarkers of cardiometabolic dysregulation, a variety of age-related chronic illnesses, and mortality. The findings are mixed, however, regarding its ability to predict dementia [53,54]. The second clock, GrimAge [55], has been shown to predict coronary heart disease, diabetes, frailty, visceral adiposity, fatty liver, cognitive decline, and time to death [56,57]; albeit, most studies failed to find an association between GrimAge and dementia [8]. The third clock, DunedinPACE, has been shown to predict physical decline, chronic illness, facial aging, and earlier mortality [58]; in addition, it has been found to predict Alzheimer’s disease and cognitive decline using a variety of markers [58,59]. These latter findings suggest that DunedinPACE assesses general biological changes that portend a decline in a wide variety of organs, including the brain. We expect to find that DunedinPACE, and possibly PhenoAge, will be related to CC-Bd, but we also expect that depression, loneliness, and BDNFm will continue to predict the acceleration of CC-Bd after the effects of these epigenetic clocks are taken into account.

To estimate the developmental trajectories of CC-Bd, we employed growth curve models with individually varying time points (age) in our analyses [60]. Previous studies investigating the correlation between cognitive decline and risk factors, such as depression, loneliness, or BDNFm, have neglected the impact of “within-person” changes and instead focused on the relationship between differences in risk factors among individuals with concurrent or delayed brain health. However, examining within-person changes can provide further clarification and support for findings from studies concentrating on between-person effects. Analyzing within-person effects enables each individual to serve as their own control, mitigating the possibility of confounding variables that are time-invariant or time-varying. Utilizing latent growth curve and parallel process models, our study expands on prior research to investigate whether within-person changes in depression, loneliness, and BDNFm are predictive of the slopes of within-person changes in CC-Bd. Even with missing data, this individual-data vector approach avoids the biased estimates of the intercept and slope obtained when participants begin a study at different ages and enables us to estimate developmental changes in the trajectories of CC-Bd from ages 30 to 70.

## 3. Materials and Methods

### 3.1. Sample

We tested our hypotheses using data collected from primary caregivers and their romantic partners in the Family and Community Health Study (FACHS), an ongoing longitudinal study of African American families initiated in 1997. The present study utilized data collected from waves 5 (2008) and 8 (2019), as these were the waves when blood samples were obtained. The FACHS sample included 889 African American families with a fifth-grade child residing in Georgia or Iowa, and the sample was designed to represent various socioeconomic statuses and neighborhood settings. At wave 1, the families were divided equally between Georgia (*n* = 422) and Iowa (*n* = 467), and the primary caregivers were predominantly women, with their partners mostly being men. At wave 5, the mean age of primary caregivers was 48.7 years (SD = 8.35), and 57.8% were married or cohabiting, whereas at wave 8, which was 11 years later, the mean age of caregivers and their romantic partners was 57.1 years (SD = 6.78), and 55.4% were married or cohabiting. The study procedures were approved by the Institutional Review Board at the University of Georgia (Title: FACHS IV; Protocol # Study00000172). Computer-assisted interviews conducted at each wave took an average of 2 h to complete.

A certified phlebotomist collected four tubes of blood (30 mL) from each consenting participant within two weeks of psychosocial interviews at wave 5 and wave 8. Eligibility for blood draws was limited to individuals still residing in Georgia or Iowa in these waves. Of the original sample, 506 participants provided blood at wave 5, and at wave 8, 480 individuals who were living in the study area and agreed to provide blood, resulting in a total sample of *N* = 693 who provided data and a blood sample at either wave 5 or wave 8. 

The Infinium MethylationEpic Beadchip (Illumina, San Diego, CA, USA) was utilized to conduct genome-wide DNA methylation evaluations in this study, which were performed by the University of Minnesota Genome Center (http://genomics.umn.edu/, accessed on 15 February 2021), following the manufacturer’s protocol. The resulting IDAT files were securely transferred, and the data DASEN normalized using the MethyLumi [61], WateRmelon [62], and IlluminaHuman MethylationEPICanno.ilm10b2.hg19 R packages, as per our previous protocols [63]. Following data normalization, sample and probe-level quality controls were conducted. Samples were removed if more than 1% of their probes had detection *p*-values of >0.05, and data from 858,924 of the 866,091 probes in the array were retained. *β*-values for each site were calculated using the standard formula where U and M are the values of the unmethylated and methylated intensity probes (averaged over bead replicates) and α = 100 is a correction term to regularize probes with low total signal intensity [64,65]. CpG values were background-corrected using the “noob” method [66]: $\beta =\frac{M}{U+M+\alpha }$. Twenty-three participants were excluded from the analysis due to missing data, leaving 670 individuals (202 men and 468 women) for the study sample.

### 3.2. Measures

#### 3.2.1. Outcome Variable

*Cortical Clock–Bd (CC-Bd)*. DNA methylation data from the Illumina EPIC 850 BeadChip (see above) was used to calculate a cortical age score for each respondent. Using an extensive dataset of human cortex tissue spanning the life course, Shireby et al. recently used elastic net regression to identify a set of 347 weighted DNA methylation sites that combined to optimally predict age in the human cortex [5]. The present study uses peripheral blood, rather than brain tissue, to calculate Shireby et al.’s epigenetic index of brain age [5]. Because the respondent’s age is controlled in our analyses, CC-Bd becomes a measure of the extent to which the participants’ brain age departs from their chronological age (i.e., it is a measure of accelerated or decelerated cortical aging).

#### 3.2.2. Predictor Variables

*Loneliness*. At Waves 5 and 8, levels of loneliness were evaluated using a subset of two items from the widely utilized UCLA Loneliness Scale [67]. Specifically, participants were asked to indicate how frequently they felt excluded and no longer close to anyone using a four-point scale ranging from 1 (never) to 4 (always). Higher scores indicate greater loneliness. Results showed a significant relationship between the two items, with a correlation coefficient of 0.43 at wave 5 and 0.59 at wave 8.

*Depression.* Depression was assessed at waves 5 and 8 using five items from The Mini Mood and Anxiety Symptom Questionnaire (mini-MASQ) [68]. Participants responded from 1 (“Not at all”) to 3 (“Extremely”) with items such as “During the past week, how much have you…felt depressed”, “felt discouraged”, “felt hopeless”, “felt like a failure” and “felt worthless”. High scores indicate greater depression. Cronbach’s α was 0.83 at Wave 5 and 0.85 at Wave 8.

*BDNFm*. The methylation of *BDNF* was assessed using Illumine EPIC 850K DNA methylation assays. Due to the correlated nature of CpG sites within *BDNF*, we employed a principal component data reduction approach to broadly characterize the level of methylation across Exon 1 and Promoter 1 of *BDNF*. We then used the first principal component extracted from these 18 sites to form our measure (see Online Supplement Appendix A). Standardized factor scores for this component were summed to form an index of *BDNFm* at both waves 5 and 8. 

*PhenoAge.* We utilized the Illumine EPIC 850K DNA methylation assays and the algorithm developed by Levine et al. [52] to calculate scores for our participants at waves 5 and 8. This algorithm is based on methylation levels at 513 CpG sites. In our study sample, the epigenetic age score had a correlation of 0.69 with chronological age. To obtain an accelerated aging score, we conducted a regression analysis where we regressed epigenetic age on chronological age. As a result, this measure of accelerated epigenetic aging is adjusted to have a correlation of 0 with chronological age. Positive values on this variable indicate accelerated epigenetic aging in years, while negative values indicate decelerated aging in years.

*Accelerated GrimAge*. At waves 5 and 8, scores were calculated for our respondents using Illumine EPIC 850K DNA methylation assays and the algorithm specified by Lu et al. [55]. This algorithm uses methylation assessments at 1030 sites across the human genome. The correlation between GrimAge and chronological age in the study sample was 0.70, which is consistent with previous studies involving other racial/ethnic groups [69]. We transformed GrimAge into an accelerated aging score by regressing it against chronological age [55]. Similar to PhenoAge, a positive score indicates accelerated epigenetic aging in years, whereas a negative score denotes decelerated aging in years.

*DunedinPACE*. The DunedinPace is a tool developed to measure physiological changes over the course of 12 months, serving as a “speedometer” for aging. If the value is greater than one, it indicates accelerated biological aging, meaning that an individual’s chronological age is advancing faster than usual. For waves 5 and 8, the DunedinPace was computed through the use of Illumine EPIC 850K DNA methylation assays and the code provided by the creators at https://github.com/danbelsky (accessed on 15 February 2021).

#### 3.2.3. Control Variables

The analyses controlled for gender, education, and chronological age.

### 3.3. Analytic Strategy

The general flow of the study is depicted graphically in Figure 1. Our statistical approach involved using growth models with time-varying covariates to test the hypotheses regarding the substantive effects of depression, loneliness, and *BDNFm* on CC-Bd. Age, centered at 30, was the measure of time in growth curve models that incorporated random effects of initial levels and linear growth rates while using individually varying observation times. All analyses were performed with Stata version 17. Missing data were handled by mixed effects models under missing at random using maximum likelihood methods [70]; this method assumes that missing data are randomly distributed and are unrelated to the dependent variable [71]. This assumption is met in the FACHS sample, as missing data are derived from the random attrition associated with a longitudinal design [72]. To correct the non-independence of the couple effects, we also included random effects for couple relationship. To simplify the models, if the variation associated with couple effects was not significant, we then dropped this random effect when we ran the conditional growth models.

The initial analysis aimed to determine the growth pattern of CC-Bd over time and its relationship with participants’ age, exploring whether changes occur consistently or accelerate with age. We then examined the effect of changes in depression, loneliness, and *BDNFm* on the variation in change in CC-Bd, controlling for age, gender, and education (steps 1 and 2 in Figure 1). Because the effect of the respondent’s age is controlled, CC-Bd becomes a measure of the extent to which the participants’ brain age departs from their chronological age. Thus, our analysis might be seen as an investigation of the extent to which changes in depression, loneliness, and *BDNFm* are associated with accelerated brain aging. Finally, we examine the extent to which the effects of our three predictors on brain age remain after introducing changes in three widely accepted biological clocks—PhenoAge, GrimAge, and DunedinPACE (Step 3 in Figure 1).

## 4. Results

The mean and standard deviation for the study variables at waves 5 (2008) and 8 (2019) are shown in Table 1. As can be seen in the table, average levels of CC-Bd increased over time and were older than chronological age at both waves. The three predictors—loneliness, depression, and BDNFm also showed increases over time, with loneliness showing the most dramatic rise. The mean loneliness score was 1.71 in 2008 versus 3.36 in 2019. GrimAge was older than chronological age at both waves, and DunedinPACE indicated that the participants went from aging biologically by 1.06 years per chronological year in 2008 to aging biologically by 1.11 years per chronological year in 2019. All of this suggests that, over time, the sample was doing more poorly emotionally and physically. These changes observed in the data necessitated the use of growth models to investigate the relationships between them.

Using an unconditional growth model with varying observation times, we investigated if there was a significant linear or nonlinear change in CC-Bd as a function of age. Analysis results showed that the best-fitting model for the data was a positive linear model, as presented in Table 2. This model had a significant effect on the linear growth rate as a function of age, as indicated by the coefficient value of b = 0.815 and a *p*-value of less than 0.01, whereas the quadratic growth model was not significant. This suggests that there was a strong relationship between the age of the subjects and their linear growth rate and that the positive linear model was an effective way to describe this relationship. In addition, there was not a significant effect associated with the random effects for the couple relationship, suggesting that partner outcomes were independent and that we could drop this term from the subsequent conditional growth models.

Turning to Table 3, Model 1 shows that at age 30, the respondents had a CC-Bd score of 38.63, and this increased by approximately 0.81 per year from age 30 to 70. The effect for gender was significant and indicated that females generally showed a lower cortical age than males. Education had no significant effect. Depression, loneliness, and BDNFm were included as time-varying covariates in Model 2. Increases in loneliness and BDNFm were positively and significantly related to linear growth in CC-Bd age, whereas the coefficient for depression did not achieve statistical significance. The Table shows that a within-person one-unit increase in loneliness was associated with a 0.638 (*p* < 0.01) year increase in cortical aging, and a within-person one-unit increase in BDNFm was associated with a 1.189 year increase in cortical aging. Model 3 shows that the significant effects of loneliness and BDNFm remain after GrimAge, DunedinPACE, and PhenoAge were included as time-varying covariates. The Table shows that two of these clocks—DunedinPACE and PhenoAge—were positively and significantly related to linear growth in cortical age. A within-person one-unit increase in DunedinePace was associated with a 2.317 year increase (*p* < 0.05) in cortical age, while a 1 year increase in PhenoAge was associated with an increase of 0.232 years (*p* < 0.01) in cortical age.

## 5. Discussion

Shireby et al. recently developed an epigenetic measure of aging based on human cortex tissue [5]. This cortical clock dramatically outperformed extant blood-based epigenetic clocks in predicting brain age and biomarkers of neurological degeneration [7]. Unfortunately, measures that require brain tissue are of limited utility to investigators striving to identify everyday risk factors for dementia. One solution is to develop clocks that target CpG sites that are conserved across the brain and other tissues. Such clocks would better represent aging processes in the brain but still be applicable across accessible tissues such as blood [7,8]. Toward that end, the present study investigated the utility of using the CpG sites included in the Cortical Clock developed by Shireby et al. [5] to formulate a peripheral blood-based cortical measure of brain age (CC-Bd) that might provide a window on brain aging that would go beyond currently available clocks.

To establish the utility of CC-Bd, we examined the extent to which three risk factors that have been linked to cognitive decline and dementia—loneliness, depression, and BDNFm—predict the acceleration of CC-Bd after controlling for age, gender, education, and three robust epigenetic clocks. Controlling for the epigenetic clocks was important as it allowed us to establish that these risk factors accelerate CC-Bd beyond any influence of general biological aging. Past research indicates that midlife assessments of each of these risk factors predict the onset of dementia as much as ten years later. We selected these three factors, in part, because there is reason to believe that they are subject to intervention. Therefore, to the extent that these risk factors predict change in CC-Bd net of the various controls, they might be seen as suggesting avenues for early intervention.

Our analysis was strengthened by utilizing growth curve modeling to estimate the predictive capability of changes in our three risk factors over time on the developmental trajectories of CC-Bd [60]. Past research regarding the association of depression, loneliness, or BDNFm with cognitive decline has not considered the impact of “within-person” change, focusing instead on the association between person differences in risk factors with concurrent or delayed brain health. Examining within-person effects allowed us to rule out time-invariant third-variables as potential confounds and enabled each person to serve as their own control. To go beyond prior work, we used latent growth curves and parallel process models to determine if within-person changes in the three predictors (depression, loneliness, and BDNFm) predict trajectories of within-person change in CC-Bd. This approach provides a more direct test of the potential of the three predictors as modifiable points of intervention to influence change in brain health.

Our findings indicated that increases in loneliness and in BDNFm were associated with the acceleration of CC-Bd relative to chronological age. Indeed, the effects were quite robust. Depression, however, was not a significant predictor of CC-Bd. Presumably, this is because of the multi-collinearity between depression and loneliness. These results suggest that, at least in the case of our African American sample, it is loneliness rather than depression that is exerting the primary effect on brain aging. The significant effects for loneliness and BDNFm held after introducing controls for three widely used epigenetic clocks. These findings suggest that CC-Bd, which uses CpG sites identified using brain tissue, is assessing something more than the pan-tissue epigenetic clocks that were trained using whole blood and considered to be measures of general biological aging. To the extent that this is the case, the results suggest that interventions to lower loneliness and increase BDNF could potentially lower a person’s risk for dementia. For example, interventions that increase social involvement might be used to reduce loneliness along with exercise and dietary changes that enhance BDNF.

It is interesting to note, however, that that within-person change over time in two of the general epigenetic clocks—Dunedin Pace and PhenoAge—was significantly related to a change in CC-BD. This is consistent with prior research showing that these two clocks predict various measures of brain pathology and cognitive decline [7,59]. Together with these prior findings, our results support the view that, at least in part, brain health is associated with the general aging of the organism. Our findings are also consistent with prior research that fails to find an association between GrimAge and either brain health or cognitive decline. This may be because it is more strongly affected by smoking and is the only clock that includes CpG sites for cigarette smoking. Given that smoking has only a very modest association with dementia and is inversely related to Parkinson’s disease [7], the strong association with smoking may suppress its association with brain aging.

In conclusion, our findings add to the literature indicating that loneliness and levels of BDNFm are important risk factors for accelerated brain aging. Further, our findings provide support for the idea that the CpG sites identified by Shireby et al. in their cortical clock based on brain tissue might be used with whole blood to construct a measure of brain age that is more accessible for observational research and superior to the pan-tissue epigenetic clocks when assessing brain aging [5]. A weakness of the present study was the absence of biomarkers of brain health. Future research needs to examine the extent to which CC-Bd shows more robust associations with biomarkers of brain pathology (e.g., Aβ load, tau tangle density, Lewy body pathology) than do the established blood-based epigenetic clocks. Grodstein et al. [7] have shown that the cortical clock using brain tissue shows much stronger correlations with brain pathology than the established blood-based clocks. It will be interesting to see if this is the case for CC-Bd as well.

## Figures and Tables

**Figure 1 genes-14-00842-f001:**
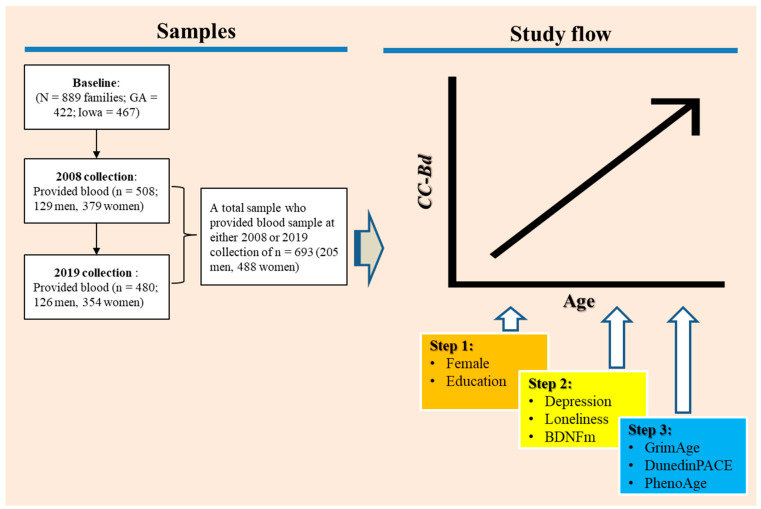
A graphical representation of the study flow.

**Table 1 genes-14-00842-t001:** Descriptive statistics for the study variables.

	2008 (*n* = 473)	2019 (*n* = 477)
Variables	Mean (SD)	Mean (SD)
Cortical Aging	50.049 (6.964)	58.027 (7.325)
Loneliness	1.705 (0.738)	3.362 (0.818)
Depression	1.242 (0.325)	1.298 (0.402)
BDNFm	−0.022 (1.965)	0.000 (1.960)
GrimAge	53.476 (8.024)	60.596 (7.153)
DunedinPACE	1.055 (0.143)	1.111 (0.141)
PhenoAge	38.236 (9.330)	46.204 (8.786)
Age	48.560 (8.226)	57.166 (6.811)
Education	12.691 (2.332)	12.979 (2.302)
Female	0.735 (0.441)	0.737 (0.440)

**Table 2 genes-14-00842-t002:** Linear and quadratic models of change in CC-Bd as a function of age using unconditional growth models (*n* = 670).

	Model 1	Model 2
Growth factor means		
Age 30	35.553 **	34.985 **
Linear growth rate	0.815 **	0.860 **
Quadratic growth rate		−0.001
Random variance		
Age 30	7.995 *	13.374 *
Linear growth rate	0.025 *	3.81 × 10^−12^
Quadratic growth rate		0.0000196 *
σ2	4.897 *	3.901 *

* *p* < 0.05 (two-tailed tests). ** *p* < 0.01

**Table 3 genes-14-00842-t003:** Parameter estimates for linear growth models using time-varying and time-invariant covariates to predict accelerated CC-Bd (*n* = 670).

	Cortical DNAm Age Clock					
	Model 1		Model 2		Model 3	
Variables	b	SE	B	SE	b	SE
Fixed Effect						
Age 30	38.630 **	1.128	37.396 **	0.114	29.812 **	1.725
Linear growth rate	0.814 **	0.016	0.685 **	0.018	0.480 **	0.033
Time-varying covariates						
Depression			0.655 †	0.367	0.463	0.337
Loneliness			0.638 **	0.106	0.573 **	0.097
BDNFm			1.189 **	0.072	1.238 **	0.068
GrimAge					−0.010	0.035
DunedinPACE					2.317 *	1.173
PheonAge					−0.232 **	0.021
Time-invariant covariates						
Female	−1.550 **	0.402	−0.923 **	0.343	−1.073 *	0.339
Education	−0.152 †	0.082	−0.059	0.070	0.021	0.066
Random effects						
τ(intercept)	7.608 *		4.952 *		3.820 *	
τ(Age)	0.024 *		0.014 *		0.013 *	
σ2	4.915 *		5.299 *		4.445 *	

Note: DunedinPACE is standardized by z-transformation (i.e., mean = 0 and SD = 1). † *p* < 0.1, * *p* < 0.05, ** *p* < 0.01 (two-tailed tests).

## Data Availability

Interested parties can contact Mei Ling Ong (tmlong@uga.edu) at the Center for Family Research at the University of Georgia to discuss access to the data used in this study.

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
