# Peer review of "Changes in Loneliness, BDNF, and Biological Aging Predict Trajectories in a Blood-Based Epigenetic Measure of Cortical Aging: A Study of Older Black Americans"

_genes, 2023, doi:10.3390/genes14040842_

Round 1

Reviewer 1 Report

To improve the perception of the material, the authors need to present the design of the experiment graphically. 

It is necessary to separate the methods of statistical data processing into a separate item and carefully describe all the procedures. 

The list of cited articles should be brought to a unified form.

In the conclusion indicate the limitations of this study.

Reviewer 2 Report

The paper entitled “ Changes in loneliness, bdnf, and biological aging predict tra- jectories in a blood-based epigenetic measure of cortical aging: A Study of Older Black Americans“ introduces further test of the extent to which CC-Bd might be functioning as a measure of brain age, independent of other epigenetic clocks.

Interesting study, solidly elaborated and discussed, however, some improvements of the results and discussion are needed.

Minor points of concern and suggestions for improvement are:

Line 331-333: please start describing the results with the important statement - not with the mean standard that belongs to the figure description.

Line 382-384: please start discussion with the summary of the most important findings.
